# Robust Federated Learning with Majority Adversaries via Projection-based Re-weighting

## Abstract

Most robust aggregators for distributed or federated learning assume that adversarial clients are the minority in the system. In contrast, this paper considers the majority adversary setting. We first show that a filtering method using a few trusted clients can defend against many standard attacks. However, a new attack called Mimic-Shift can circumvent simple filtering. To this end, we develop a re-weighting strategy that identifies and down-weights the potential adversaries under the majority adversary regime. We show that our aggregator converges to a neighborhood around the optimum under the Mimic-Shift attack. Empirical results further show that our aggregator achieves negligible accuracy loss with a majority of adversarial clients, outperforming strong baselines.

## 1 Introduction

Federated learning (FL) is a leading framework for collaboratively training a machine learning (ML) model over local datasets. The decentralized nature of FL systems has raised concerns about vulnerability – as adversaries can connect to an FL system like other benign users and corrupt the ML model while evading detection by standard means (Kairouz et al., 2021). To this end, there is growing literature on the adversarial robustness of FL (Blanchard et al., 2017; Chen et al., 2018; Xie et al., 2019b; Rajput et al., 2019b; Xie et al., 2020; Karimireddy et al., 2021a; 2022; He et al., 2022b), particularly where adversaries can upload malicious updates. Most existing defenses assume that the adversarial clients are the minority in the system (Blanchard et al., 2017; Chen et al., 2018; Rajput et al., 2019b; Karimireddy et al., 2021a; He et al., 2022b). However, in a federated scenario, the decentralized nature means that it is relatively straightforward for the adversary to be the majority and thus break existing defenses. We call such an adversary the "majority adversary".

Our work joins a growing literature on robustness with majority adversaries, e.g., Xie et al. (2019b; 2020), motivated by noted practical vulnerabilities. Although Shejwalkar et al. (2021) argue that the number of registered clients in a production system (e.g., GBoard) may be too large for the adversary to compromise a majority of them, they neglect the client availability issue in FL. In particular, Kairouz et al. (2021) suggests that, at any given time, only a subset ($< 1\%$) of clients are available for the server. Such a low client availability allows the adversary to become the majority and overwhelm the server utilizing compromised networked devices (e.g., IoT devices) in a similar way as the common distributed denial-of-service (DDoS) attack (Specht & Lee, 2003; Bonguet & Bellaïche, 2017). Some other settings such as crowd-sourced training (Ryabinin & Gusev, 2020) (a.k.a. volunteer computing) are perhaps even more vulnerable to majority adversaries because the crowd-sourcing systems do not implement access control – allowing the adversary to connect an arbitrary number of clients as volunteers.

We consider the adversarial robustness of federated learning against a class of attacks where an adversary aims to decrease the accuracy of the trained ML model by uploading malicious updates. In particular, we are interested in Mimic-type attacks (Karimireddy et al., 2022), as is discussed later in this section. A key assumption in our setup is the existence of a few trusted clients, e.g., with secure hardware support. We call these trusted clients "reference clients". In practice, the number of reference clients could be as small as two in each round. Similar approaches have been considered in existing works (Xie et al., 2019b; 2020). One option for secure hardware is the trusted execution environment (TEE) (Pinto & Santos, 2019), which guarantees that the program is not Byzantine. TEE is so far commercialized (e.g., on Google Pixel (GoogleBlog), Apple iPhone (AppleSupport), Sam-

sung phones (SamsungDeveloper)). Recent works (Mo et al., 2021) bring TEE support to federated learning systems.

We propose a combination of defenses – filtering and projection-based re-weighting. Our first defense is a filtering method that constructs a spherical accept region and excludes the updates outside it. The center of the accept region is an average update from a few trusted clients with secure hardware support. The radius of the accept region is the sample variance of reference updates times a scaling factor. Although the filtering method is effective against several standard attacks (e.g., sign flipping attack, Gaussian attack), this filtering is easy to circumvent.

Building on the recently proposed Mimic attack (Karimireddy et al., 2022), shown to break many existing defenses, we develop an improved attack method called Mimic-Shift. Mimic-Shift clients send malicious updates which slightly shift away from benign updates and mislead the aggregated updates away from the expected update. The slight shift makes Mimic-Shift hard to detect, and the calibrated shifting direction can corrupt the aggregated model. To perform the Mimic-Shift attack, we consider a man-in-the-middle (MITM) adversary capable of intercepting the message between the clients and the server. Computer security researchers have extensively studied the MITM adversary, but existing encryption solutions can be too expensive for resource-constrained client devices. For example, the AES (advanced encryption standard) encryption only has a throughput of around 50 MB/s even with a powerful desktop CPU (Gleeson et al., 2014). With a 1 GB moderate size neural network, the encryption takes more than 20 seconds on the client-side, draining the computational resources and increasing the client dropout rate.

Our second defense is a projection-based re-weighting method to deal with the Mimic-Shift attack under a majority adversary regime. The main idea is to measure the influence of each update on the aggregated update, then down-weight the updates with high influence. Specifically, we compute the scalar projection of the aggregated update on each client's update. The intuition is that the majority adversarial clients can significantly mislead the aggregated update, resulting in a large scalar projection. Note that the filtering and re-weighting methods complement each other because the re-weighting defense does not deal with the aforementioned standard attacks, as is discussed in Section 4.

We further provide some theoretical analysis of our methods under the Mimic-Shift attack. First, false-positives in filtering can eliminate a benign update and perturb the aggregated update. Regarding this concern, we show that the false positive rate decreases quickly w.r.t. the number of reference clients and the scaling factor of the accept radius, suggesting that our filtering method can work with a few reference clients and a conservative scaling factor. For the re-weighting phase, the probability of malicious updates having larger scalar projections than benign updates increases as the adversary takes more shares in a system, complementing existing results (He et al., 2022b) that guarantee more robustness as the share of the adversary decreases. Additionally, we discuss the performance of our method under a conventional minority adversary setting. Finally, we show that, in a convex setting, our method converges with a rate of $\mathcal{O}(\frac{1}{\sqrt{T}})$ to a neighborhood of the optimum. Our contributions are summarized as follows:

- We develop the Mimic-Shift attack and show that Mimic-Shift circumvents many defense methods in federated learning.

- We develop a two-stage defense using filtering and re-weighting to defend against a broad class of attacks.

- We theoretically analyze our strategy and outline conditions under which it helps.

Empirical results on FEMNIST (Caldas et al., 2018), CelebA (Liu et al., 2015) and Shakespeare (McMahan et al., 2017) datasets show that our aggregator recovers a near-optimal model under a majority adversary setting with Mimic-Shift attack, outperforming existing methods by a large margin. Also, our method only loses up to 2.4% accuracy under conventional minority adversary settings. Additional empirical results demonstrate that our method is robust to a broad class of attacks, including standard Gaussian and sign-flipping attacks as well as an improved Mimic-Shift-Var attack.

## 2 RELATED WORK

Many existing works on Byzantine robust machine learning assume the adversary is the minority in the system, based on clustering (Blanchard et al., 2017), median (Yin et al., 2018), voting (Bernstein et al., 2019), bucketing (Karimireddy et al., 2022), and robust estimation (Data & Diggavi, 2021). However, these methods can fail once the adversary becomes the majority. A few papers (Xie et al., 2019b; 2020) leverage a validation dataset to improve Byzantine tolerance to arbitrary numbers of adversarial clients but constructing a global validation dataset in a federated scenario with local private datasets is infeasible. Other redundancy-based methods with more robustness guarantees do not apply to federated learning because they assume a centralized setting with control over the data allocation on each node (Chen et al., 2018; Rajput et al., 2019a). Recently, a clipping-based method (Karimireddy et al., 2021b; He et al., 2022b) showed success against minority adversary in a federated learning setting. Although it is relatively straightforward to combine the clipping with our reference clients, we find that the training is unstable, as is discussed in Section 6.

Our paper focuses on model poisoning attacks, which aim to decrease the accuracy of the trained ML model by uploading malicious updates (He et al., 2022b). Other attacks, including data poisoning (Steinhardt et al., 2017; Wang et al., 2021) and backdoor (Wang et al., 2020; Xie et al., 2021), are beyond the scope of this work. We focus only on training the centralized model. Additional considerations such as personalization are known to be handled effectively using robust centralized training, followed by local fine-tuning Li et al. (2021), thus are beyond the scope of this paper.

## 3 PROBLEM SETUP

We assume a federated learning system where $N$ benign clients collaboratively train a ML model $f : \mathcal{X} \to \mathcal{Y}$ with $d$-dimensional parameter $\zeta$ coordinated by a server. The $N$ benign clients include $N_R$ reference (trusted) clients. There are $N'$ adversarial clients who aim to corrupt the ML model during training. The $i^{\text{th}}$, $i \in [1, ..., N + N']$, client has $n_i$ data samples, being benign for $i \in [1, ..., N]$ or being adversarial for $i \in [N + 1, ..., N + N']$. The federated learning is conducted in $T$ rounds. In round $t \in [1, ..., T]$, the server broadcasts a model parameterized by $\zeta_{t-1}$ to each client. We omit the subscript $t$ while focusing on one round. Then, the $i^{\text{th}}$ client optimizes $\zeta_{t-1}$ with their local data samples and report $\zeta_{t,i}$ to the server. We define pseudo-gradient $g_{t,i} = \zeta_{t-1} - \zeta_{t,i}$ being the difference between the locally optimized model and the broadcasted model from the previous round. Note, for simplicity, that we will often use the term "gradient" to refer to the updates. Once all the gradients are reported in, the server aggregates the gradients and produce a new model with parameters $\zeta_t$ using the following rule: $\zeta_t = \zeta_{t-1} - \sum_{i=1}^{N+N'} \frac{n_i}{\sum_{i=1}^{N+N'} n_i} g_{t,i}$.

Our goal is to minimize a risk function over the benign clients: $F(\zeta) = \sum_{i=1}^{N} \frac{n_i}{\sum_{i=1}^{N} n_i} F_i(\zeta) = \sum_{i=1}^{N} \frac{n_i}{\sum_{i=1}^{N} n_i} \mathbb{E}_{\mathcal{D}_i}[\ell(f(x; \zeta), y)]$, where $\ell : \mathbb{R} \times \mathcal{Y} \to \mathbb{R}$ is a loss function.

### 3.1 THREAT MODEL

Following the standard practice (Blanchard et al., 2017), we consider an omniscient man-in-the-middle (MITM) adversary that knows $g_i, \forall i \in \{1, ..., N\}$ and can tell which clients are reference users, e.g., by observing the device identifier in the message. An omniscient adversary helps explore the limits of the defense. However, an omniscient adversary is not mandatory for our Mimic-Shift attack, which can work with partial information as Section 6 will show. The MITM adversary also owns a majority of clients in a system and adopts an attack from the Mimic family (Karimireddy et al., 2022). Specifically, we assume the following Mimic-Shift attack.

**Mimic-Shift**    At each round, the MITM adversary first intercepts the gradients from benign clients. Then the adversary computes the average reference gradient $\bar{g}_R = \sum_{i=1}^{N_R} \frac{n_{R_i}}{\sum_{i=1}^{N_R} n_{R_i}} \cdot g_{R_i}$ from the reference users indexed by $R_i$ and the average benign gradient $\bar{g} = \sum_{i=1}^{N} \frac{n_i}{\sum_{i=1}^{N} n_i} \cdot g_i$ from all benign users, including the reference users. Then, all the adversarial clients report $g' = \bar{g}_R + (\bar{g}_R - \bar{g})$ to the server.

The Mimic-Shift attack is effective because it tries to push the aggregated gradient away from the expected gradient. The reason is that $\bar{g}$ is estimated with more clients and data samples. A simple concentration argument suffices to show that $\bar{g}$ will be closer to $\mathbb{E}[\bar{g}]$ than $\bar{g}_R$ with high probability,

resulting in a $\bar{g}_R - \bar{g}$ pointing away from $\mathbb{E}[\bar{g}]$. Empirical results in Section 6 show that Mimic-Shift decreases the accuracy twice as much as Mimic (Karimireddy et al., 2022), which sets $g' = g_i$ for an $i \in \{1, ..., N\}$.

The Mimic-Shift attack is also difficult to detect because the malicious $g'$ has the same distance to the reference $\bar{g}_R$ as the benign $\bar{g}$. We will see how this property lets Mimic-Shift bypass distance-based defenses in Section 6.

## 4 METHOD

Our robust aggregator has two phases, filtering and re-weighting. The filtering phase, applied first, excludes the gradients outside the accept region defined by the reference gradients, defending against standard attacks. Then, the re-weighting phase further down-weights the potential malicious gradients within the accepting region of the filtering phase, targeting the Mimic-Shift attack.

### 4.1 PHASE 1: FILTERING

In the filtering phase (Algorithm 1), we first compute the sample mean of reference gradient $m_R = \sum_{i=1}^{N_R} \frac{1}{N_R} \cdot g_{R_i}$, making a guess of where the good gradients might be. Next, we set $m_R$ as the center of a spherical accept region. Then, we compute the sample variance of the reference gradients, $s_R = \frac{\sum_{i=1}^{N_R} \|g_{R_i} - m_R\|}{N_R - 1}$. The sample variance $s_R$ is further scaled up by a tunable hyper-parameter $c$. The product $c \cdot s_R$ is the radius of the spherical accept region. Finally, we remove all the updates outside the spherical accept region. This filtering is effective against attacks that do not know where the expected gradient is (e.g., Gaussian attack) or do not carefully calibrate the adversarial gradient (e.g., sign flipping attack), as is shown in Section 6.

---

**Algorithm 1** Filtering.

---

**Input:**
   A set of reference gradients, $\{g_{R_i} \mid i \in \{1, ..., N_R\}\}$;
   A set of reported gradients, $\{g_i \mid i \in \{1, ..., N + N'\}\}$;
   A hyper-parameter $c$;

**Aggregator:**
1: Compute the sample mean of reference gradients, $m_R := \sum_{i=1}^{N_R} \frac{1}{N_R} \cdot g_{R_i}$;
2: Compute the sample variance of reference gradients, $s_R := \frac{\sum_{i=1}^{N_R} \|g_{R_i} - m_R\|}{N_R - 1}$;
3: **return** $\{g_i \mid i \in \{1, ..., N + N'\} \wedge \|g_i - m_R\| \leq c \cdot s_R\}$;

---

### 4.2 PHASE 2: RE-WEIGHTING

Suppose a majority adversary misleads the aggregated gradient. In that case, the malicious gradients likely have a stronger influence on the aggregated gradient compared to benign gradients. Here, we measure the influence via scalar projections.

The re-weighting defense is outlined in Algorithm 2. The scaling weights are designed to augment benign gradients and down-weight adversarial updates. This is implemented using a monotonic re-scaling [1] of the scalar projection between the aggregate and the gradients (step 4). The monotonic re-scaling amplifies the difference between the scalar projections, making large projections larger. Therefore, the potential malicious gradients would be down-weighted more. However, the monotonic re-scaling may also lead to an over-up-weighting on certain benign gradients, which are far from $\bar{g}^*$. such up-weighting leads to instability in the training process. Thus, we also include a clipping operator (step 6). The clipping addresses the over-up-weighting issue by specifying a bound.

An additional benefit of using the power function and clipping is that they prompt a set of uniform weights on benign gradients, stabilizing the training. If we set $k$ to be sufficiently large such that the maximum $s_i$ is greater than $N_F \cdot (\sum_{i=1}^{N_F} s_i - \max_{i \in \{1, ..., N_F\}} s_i)$, we have $s_i = \tau, \forall i \neq$

---

[1]In the description and experiments, we use a power function. Optimizing the choice of monotonic re-scaling is left for future work.

$\arg \max_{i \in \{1, ..., N_F\}} s_i$. In practice, we set $k$ to 10 and $\tau$ to 0.6. Both hyper-parameters generalize well to the three datasets in our experiments.

Note that the filtering phase complements the re-weighting phase because standard attacks can corrupt the scalar projections via uploading 0 or flipping the sign. For a class of inner product-based attacks (Xie et al., 2019a), an additional clipping operator that prevent the scalar projection $s_i$ from being negative can help.

---

**Algorithm 2** Re-weighting.

---

**Input:**
    A set of filtered gradients, i.e., the results of Algorithm 1, $\{g_i \mid i \in \{1, ..., N_F\}\}$;
    Two hyper-parameters $k$ and $\tau$;
**Aggregator:**
 1: Compute the aggregated gradients of all users, $\bar{g}^* := \sum_{i=1}^{N_F} \frac{n_i}{\sum_{i=1}^{N_F} n_i} g_i$;
 2: Compute a scalar projection of $\bar{g}^*$ on each $g_i$, $s_i := \frac{\bar{g}^* \cdot g_i}{\|g_i\|}$;
 3: Normalize the vector $s = [s_1, ..., s_{N_F}]$ such that $\|s\|_1 = N_F$;
 4: Take the $k^{\text{th}}$ power of each $s_i$, $s := [s_1^k, ..., s_{N_F}^k]$;
 5: Normalize the vector $s = [s_1, ..., s_{N_F}]$ such that $\|s\|_1 = N_F$;
 6: Clip the values smaller than $\tau$ in s, $s_i := \max(s_i, \tau), \forall i \in \{1, ..., N_F\}$;
 7: Normalize the vector $s = [s_1, ..., s_{N_F}]$ such that $\|s\|_1 = N_F$;
 8: Re-weight each $g_i$ using $s_i$, $g_i := \frac{g_i}{s_i}, \forall i \in \{1, ..., N_F\}$;
 9: **return** $\bar{g}j := \sum_{i=1}^{N_F} \frac{n_i}{\sum_{i=1}^{N_F} n_i} g_i$;

---

## 5 THEORETICAL ANALYSIS

We first show that the probability of not filtering out a benign gradient increases at a rate of $\mathcal{O}(1 - \frac{1}{N_R^2})$ w.r.t. the number of reference clients $N_R$ and $\mathcal{O}\left((1 - \frac{1}{c^2})^2\right)$ w.r.t. the scaling factor $c$ in the accept radius. Then, we discuss conditions for the probability of down-weighting the malicious gradients to increase as the adversary owns more clients in the system. Additional discussion considers how our strategy may still help even if the aforementioned conditions do not hold, providing insights to certain of empirical results. Finally, we study the convergence of our method under a convex setting. Before proceeding, we outline some additional assumptions.

To simplify tedious notation, we assume all users have the same number of samples because the difference in sample size can be merged into the difference between gradients.

**Assumption 1.** *Uniform sample size:*

$$n_i = n_j, \forall i \neq j. \tag{1}$$

**Assumption 2.** *The gradients follow a hierarchical distribution in all rounds:*

$$\begin{aligned} &\text{Stage 1}: \mu_i \sim P(\mu), \sigma_i \sim P(\sigma), \forall i \in \{1, ..., N\}, \\ &\text{Stage 2}: g_i \sim \mathcal{N}(\mu_i, \sigma_i; \gamma^+), \forall i \in \{1, ..., N\}, \end{aligned} \tag{2}$$

*where Stage 1 is a distribution where the random variable $\mu$ has finite expected value $\mathbb{E}[\mu]$ and finite non-zero variance $\text{Var}[\mu]$. Stage 2 is truncated isotropic Gaussian distribution with a truncation threshold $\gamma^+$ on the $L_2$ norm.*

**Assumption 3.** *The variance of gradients is bounded:*

$$\sigma_i \leq \sigma^+, \forall i \in \{0, 1, ..., N\}. \tag{3}$$

**Assumption 4.** *The sample variance of reference gradients is greater than 0:*

$$0 < s_R^- \leq s_R. \tag{4}$$

**Assumption 5.** *The $L_2$ norm of expected and estimated gradients is bounded:*

$$\begin{aligned} 0 &< \gamma^- \leq \|\mu_i\| \leq \gamma^+, \forall i \in \{0, 1, ..., N\}, \\ 0 &< \gamma^- \leq \|g_i\| \leq \gamma^+, \forall i \in \{0, 1, ..., N\}. \end{aligned} \tag{5}$$

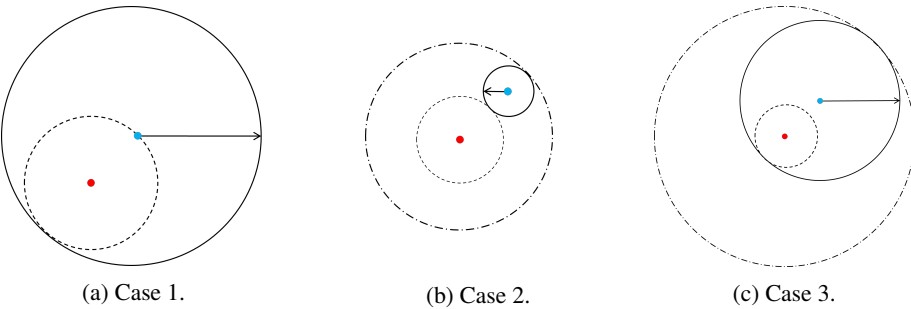

(a) Case 1.        (b) Case 2.        (c) Case 3.

Figure 1: Three cases with different hyper-parameters $c$. The red dot is $\mathbb{E}[\mu]$ and the blue dot is $\bar{\mu}_R$. The solid circle represents the accept region with radius $c \cdot s_R$. The dash circle and dash dot circle(s) are two spherical contours around $\mathbb{E}[\mu]$. Condition $c \geq 2 \cdot \frac{\|\mathbb{E}[\mu] - \bar{\mu}\|}{s_R}$ holds in Case 1, where all the points outside the accept region are more than $\frac{c \cdot s_R}{2}$ away from $\mathbb{E}[\mu]$. In Case 2 and Case 3, there are points between the two contours that have the same distance to $\mathbb{E}[\mu]$ but are not always inside accept region.

## 5.1 FILTERING (STAGE 1)

The filtering phase removes all the gradients that are more than $c \cdot s_R$ away from the reference $m_R$ in terms of Euclidean distance. However, the data distribution $\mathcal{D}_i$ may vary across clients in a federated learning setting. Such non-i.i.d.ness increases the risk of filtering out a benign gradient. We start our analysis by considering the expected gradients:

**Lemma 6.** *Suppose there are $N_R$ reference clients among $N$ clients, and the sample variance of gradients is $s_R$, under Assumption 2, let $c \geq 2 \cdot \frac{\|\mathbb{E}[\mu] - \bar{\mu}\|}{s_R}$. Then, for any $i \in [1, ..., N]$, with probability at most $\frac{4 \cdot \mathrm{Var}[\mu]^2}{c^2 \cdot s_R^2}$, we have $\|\mu_i - \bar{\mu}_R\| \geq c \cdot s_R$, where $\bar{\mu}_R = \sum_{i=1}^{R} \frac{1}{N_R} \cdot \mu_{R_i}$.*

**Remark 7.** *Condition $c \geq 2 \cdot \frac{\|\mathbb{E}[\mu] - \bar{\mu}\|}{\mathrm{Var}[\mu]}$ guarantees that all $\mu_i$s outside the accept region are at least $\frac{c \cdot s_R}{2}$ away from $\mathbb{E}[\mu]$, as Figure 1 shows, enabling concentration argument. Otherwise, bounding the probability of $\mu_i$ appears outside the accept region is hard without additional assumptions on $P(\mu)$.*

Lemma 6 suggests that the probability of filtering out a benign gradients decreases at a rate of $\mathcal{O}(\frac{1}{c^2})$. The following lemma further shows the probability of violating the assumed condition $c \geq 2 \cdot \frac{\|\mathbb{E}[\mu] - \bar{\mu}\|}{s_R}$ decreases at a rate of $\mathcal{O}(\frac{1}{N_R^2})$.

**Lemma 8.** *With Assumptions 2 and 5, for a fixed accept radius $c \cdot s_R$, with probability at most $\frac{4 \cdot \mathrm{Var}[\mu]^2}{N_R^2 \cdot c^2 \cdot s_R^2}$, we have $c \leq 2 \cdot \frac{\|\mathbb{E}[\mu] - \bar{\mu}\|}{s_R}$.*

The following theorem combines and extends the result on expected gradients to estimated gradients.

**Theorem 9.** *Under Assumptions 2 and 3, with probability at least $\frac{1}{(2 \cdot \sigma^+ \cdot \sqrt{2\pi})^d} \cdot (1 - \frac{4 \cdot \mathrm{Var}[\mu]^2}{c^2 \cdot s_R^2}) \cdot (1 - \frac{4 \cdot \mathrm{Var}[\mu]^2}{N_R^2 \cdot c^2 \cdot s_R^2})$, we have $\|g_i - m_R\| \leq c \cdot s_R$.*

Theorem 9 shows that the risk of filtering out a benign gradient decreases quickly w.r.t. $c$ and $N_R$, enabling our strategy of using a small number of reference clients and a conservative $c$.

## 5.2 RE-WEIGHTING (STAGE 2)

Suppose the adversary adopts the Mimic-Shift attack and bypasses the filtering defense. We would like to figure out the probability of assigning a higher scalar projection to the malicious gradient $g'$ than a benign gradient $g_i, i \in \{1, ..., N\}$.

**Theorem 10.** *Suppose all the gradients pass the filtering phase. Let the aggregated gradient be $\bar{g}^* = w \cdot \bar{g} + w' \cdot g'$, where $(w, w') = \left( \frac{\sum_{i=1}^{N} n_i}{\sum_{i=1}^{N+N'} n_i}, \frac{\sum_{i=N+1}^{N+N'} n_i}{\sum_{i=1}^{N+N'} n_i} \right)$. Let $\theta$ be the angle between $\bar{g}$ and*

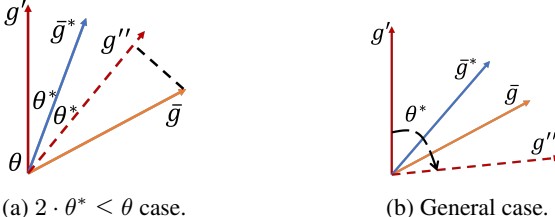

(a) $2 \cdot \theta^* \leq \theta$ case.  (b) General case.

Figure 2: Two cases where the adversary mislead the aggregated gradient $\bar{g}^*$ by different degree. $\bar{g}$ is benign. $g''$ is a mirror of malicious $g'$ w.r.t. $\bar{g}^*$, along which the scalar projection equals to $s'$.

$g'$, $\theta^*$ be the angle between $\bar{g}^*$ and $g'$. Assume $\frac{w}{w'} \leq \frac{\|g'\| - \cos\frac{\theta}{2} \cdot \|\bar{g}\|}{\|\bar{g}\| - \cos\frac{\theta}{2} \cdot \|g'\|}$ such that $\theta^* \leq \theta - \theta^*$. Then, under Assumptions 1- 5, let $\gamma_{\theta^*}^- = \|\gamma^-\| \cdot \tanh(\theta - 2 \cdot \theta^*)$, with probability at least $\frac{1}{(2 \cdot \sigma^+ \cdot \sqrt{2\pi})^d}$ · $\int_{\alpha=0-}^{\gamma_{\theta^*}^-} \left(1 - \frac{\mathrm{Var}[\mu]}{\alpha^2}\right) \cdot \left(1 - \frac{\mathrm{Var}[\mu]}{(\gamma_{\theta^*}^- - \alpha)^2}\right) d\alpha$, we have $s' \geq s_i, \forall i \in \{1, ..., N\}$.

Theorem 10 suggests that if $2 \cdot \theta^* \leq \theta$, the probability of down-weighting malicious gradients more than benign gradients (i.e., $s' > s_i$) increases as the adversary owns more clients and samples in a system (i.e. $w' \uparrow$). The main idea is: the minimum amount of deviation between $g_i$ and $\bar{g}$ for having $s_i > s'$ increases as $w'$ increases because $w' \uparrow \rightsquigarrow \theta^* \downarrow$. This specific deviation is shown in Figure 2a as the dark dash line with length at least $\|\gamma^-\| \cdot \tanh(\theta - 2 \cdot \theta^*)$. Then, applying concentration arguments yields the result.

### 5.2.1 Additional Discussion

Theorem 10 only deals with the case where $\bar{g}^*$ has a smaller angle with $g'$ than $\bar{g}$. However, in practice, we find that our method also achieves a decent accuracy even the condition $2 \cdot \theta^* \leq \theta$ does not hold. To gain some insight, suppose there is a probability density function $f$ of benign gradients. Then, the probability of having $s_i > s', \forall i \in \{1, ..., N\}$ equals the integral of $f$ over a cone, which is defined by spinning $g'$ around $\bar{g}^*$ as Figure 2b shows. In a federated scenario, the estimated $g_i$ concentrates around its own $\mu_i$, which is not necessarily close to $\bar{g}$ or $\bar{\mu}$, resulting in a small integral of $f$ over the aforementioned cone.

### 5.3 Convergence Analysis

So far, our analysis has focused on the robustness of one update. The following extends the single step analysis to a convergence analysis. Before showing the theorem, we present a lemma that quantifies the impact of false-positive filtering.

**Lemma 11.** *Suppose $\mathcal{F} : N \times R^d \to N \times \{0, 1\}$ is a filtering function, let mask $M = \mathcal{F}(g_1, ..., g_N)$, $N'_F = 1 - \|M\|_1$, $\hat{g} = \frac{1}{\|M\|_1} \cdot \sum_{i=1}^{N} M_i \cdot g_i$, and $\delta = \hat{g} - \bar{g}$. Then, with Assumption 5, we have $\|\delta\| \leq \frac{2 \cdot N'_F}{N} \cdot \gamma^+$.*

In the following theorem, we use assumed conditions to ease the reading. Later , we shall connect the assumed conditions to Theorems 9 and 10 and further support the assumed conditions with empirical results in Appendix C.

**Theorem 12.** *Suppose the malicious gradient $g'_t$ is filtered out with probability 0, $\forall t \in \{1, ..., T\}$, and down-weighted by $s'_t \geq 1$. Assume at most $N'_F$ benign gradient are filtered out at each iteration. Define an aggregated gradient $\bar{g}^*_t = w \cdot \hat{g}_t + w' \cdot \frac{g'_t}{s'_t} = w \cdot (\bar{\mu}_t + \delta_t + \epsilon_t) + w' \cdot \frac{g'_t}{s'_t}$, where $\hat{g}_t$ and $\delta_t$ follow the definitions in Lemma 11, $\epsilon_t \sim \frac{1}{N} \cdot \sum_{i=1}^{N} \mathcal{N}(0, \sigma_{t,i})$ as Assumption 2 shows. Let $\zeta \in Z$ be the model parameter, $\zeta^*$ be the optimum, and $\bar{\zeta} = \frac{1}{T} \sum_{t=1}^{T} \zeta_t$. Assume $\sup_{\zeta \in Z} \|\zeta\| \leq D$ and $F : Z \to \mathbb{R}$ is convex. Let $C = w \cdot \frac{2 \cdot N'_F}{N} \cdot \gamma^{+2} + 2 \cdot w \cdot \gamma^{+2} + w' \cdot \gamma^{+2}$, $C' = 2 \cdot w \cdot C + C^2 + w \cdot \gamma^+$, and $\eta_t = \frac{D}{\sqrt{T} \cdot C'}$, under Assumptions 2 and 5, we have:*

$$\mathbb{E}[F(\bar{\zeta})] - F(\zeta^*) \leq \frac{(4 \cdot C' + 1) \cdot D}{2 \cdot w \cdot \sqrt{T}} + \frac{2 \cdot D}{T} \cdot \sum_{t=1}^{T} \left(\|\delta_t\| + \frac{w'}{w \cdot s'_t} \cdot \|g'_t\| + \|\epsilon_t\|\right). \quad (6)$$

The convergence analysis shows that our method converges to a neighborhood around the optimum, whose size shrinks with fewer false-positive filtering (Theorem 9) and more down-weighting (tuning $k$ and $\tau$ in Algorithm 2) with higher probability (Theorem 10) on the malicious gradients. Converging to a neighborhood is common in previous noisy gradient descent studies (Wang et al., 2021; He et al., 2022a). Our result also suggests scaling up the learning rate as the adversary owns more clients. In practice, this scaling is handled by our re-weighting phase where the benign gradients are down-weighted by scalars less than 1.

## 6 EXPERIMENTS

We first compare the Mimic-Shift attack with the Mimic attack, showing that Mimic-Shift is more effective and the improved effectiveness remains with a non-omniscient adversary (Mimic-Shift-Par). Then, we evaluate our defense strategy against the Mimic-Shift attack where our strategy outperforms strong baselines.

**Additional Results**  Appendix C provides more results. We evaluate our defense via a non-omniscient Mimic-Shift-Par attack, an improved Mimic-Shift-Var attack as well as other standard attacks (e.g., Gaussian and sign-flipping). There are also additional experiment with fewer clients and various attack strengths, plots of the re-weighting vector $s$ as well as convergence, and studies on defense hyper-parameters.

### 6.1 SETUP

We use three datasets, FEMNIST (FM) (Caldas et al., 2018), CelebA (CA) (Liu et al., 2015), and Shakespeare (SS) (McMahan et al., 2017), with realistic non-i.i.d. partitions (Caldas et al., 2018) implemented in the FedML library (He et al., 2020). The number of benign clients ranges from 143 to 500. We select 2 reference clients and 28 other benign clients at each round. The number of adversarial clients is adjusted accordingly. We report the detailed setup in Appendix B, including various hyper-parameters.

We employ seven baselines, including federated averaging (FedAvg) (McMahan et al., 2017), federated averaging with reference clients (Ref), coordinate-wise median (CM) (Yin et al., 2018), FLTrust (Cao et al., 2021), Krum (Blanchard et al., 2017), Krum with bucketing (Krum-B) (Karimireddy et al., 2022), self-centered clipping with reference clients (CClip-R) (Karimireddy et al., 2022; He et al., 2022b), Zeno′ (Xie et al., 2019b), and FLTrust (Appendix C.1) (Cao et al., 2021). Appendix B lists the details of these baselines. The "Oracle" aggregator operates with benign gradients and serves as a reference. Five attacks are considered in our experiments, including Mimic-Shift-Var, particularly designed as a strong attack for our proposed defense.

**Gaussian**  Draw a random update $g_i'$ from an isotropic Gaussian distribution $\mathcal{N}(0, 200)$.

**Sign-flipping**  Flip the sign of the estimated gradient $g_i' = -g_i$ and report $g_i'$ to the server.

**Mimic-Shift**  Report $g' = \bar{g}_R + (\bar{g}_R - \bar{g})$ to the server, as is shown in Section 3.1.

**Mimic-Shift-Par**  Randomly eavesdrop 20% clients per round and draw two clients as reference.

**Mimic-Shift-Var**  Mirror the local update using $\bar{g}_R$ and report $g_i' = \bar{g}_R + (\bar{g}_R - \bar{g}_i)$ to the server.

### 6.2 MIMIC-TYPE ATTACK COMPARISON

Table 1 shows the accuracy of FedAvg aggregator under Mimic and Mimic-Shift attack. Here, the adversary owns 80% of the system, and the Mimic adversary mimics the first client in the system. Mimic-Shift attack constantly outperforms Mimic and is 116% more effective on average. Such advantages are preserved under a non-omniscient adversary.

Table 1: Accuracy of FedAvg Aggregator under Attack with 80% Adversary

| Attack Method | FEMNIST | CelebA | Shakespeare | Avg. Decrease |
|---|---|---|---|---|
| No Attack | $.861 \pm _{.001}$ (.000↓) | $.869 \pm _{.001}$ (.000↓) | $.364 \pm _{.001}$ (.000↓) | .000↓ |
| Mimic | $.693 \pm _{.001}$ (.168↓) | $.816 \pm _{.001}$ (.053↓) | $.345 \pm _{.001}$ (.019↓) | .078↓ |
| Mimic-Shift | $.621 \pm _{.001}$ (.240↓) | $.797 \pm _{.001}$ (.072↓) | $.169 \pm _{.001}$ (.195↓) | .169↓ |
| Mimic-Shift-Par | $.663 \pm _{.001}$ (.198↓) | $.812 \pm _{.001}$ (.054↓) | $.182 \pm _{.001}$ (.182↓) | .145↓ |

Note: Variance is rounded up.

## 6.3 DEFENSE AGAINST MIMIC-SHIFT ATTACK

This experiment considers three settings where the adversary owns 0% - 80% of a system. The hyper-parameters of each aggregator are selected based on the 80% adversary setting and directly applied to the other settings. Table 2 shows the accuracy of our strategy and baselines under the Mimic-Shift attack.

Under a majority adversary setting, our method outperforms all the baselines by a large margin. The reason is that the median-based (CM) method picks the malicious gradient once the adversary becomes the majority. Clustering-based (Krum and Krum-B) methods suffers from a similar issue. Other reference client-assisted distance-based re-weighting strategy (CClip-R) and filtering strategy (Zeno′) is not effective because Mimic-Shift carefully calibrates the malicious gradients so that they are as close to the reference gradients as the benign gradients. Using only reference clients (Ref) outperforms existing robust aggregators but suffers from low client utilization.

We also find that our proposed defense yields the best accuracy under a minority adversary setting, as discussed in Section 5.2.1. Under a no adversary setting, our strategy weights the benign gradients nearly uniformly without interfering with the training. Our method occasionally outperforms the oracle. We hypothesize that the improved results can come from the up-weighting of the under-represented clients (Li et al., 2020), whose influence score is small.

Table 2: Accuracy of Aggregators Under Mimic-Shift Attack

| Adv % | Data | Oracle | Ours | FedAvg | Ref | CM | Krum | Krum-B | CClip-R | Zeno′ |
|---|---|---|---|---|---|---|---|---|---|---|
| 80% | FM | .861 | **.840** | .621 | .761 | .535 | .554 | .525 | .562 | .527 |
| | CA | .870 | **.877** | .797 | .820 | .782 | .787 | .865 | .792 | .805 |
| | SS | .364 | **.360** | .169 | .236 | .189 | .186 | .183 | .160 | .194 |
| | Avg | .698 | **.692** | .529 | .606 | .502 | .509 | .525 | .505 | .509 |
| 40% | FM | .861 | **.871** | .797 | .761 | .815 | .537 | .603 | .583 | .533 |
| | CA | .870 | **.893** | .862 | .820 | .857 | .836 | .858 | .839 | .845 |
| | SS | .364 | **.340** | .291 | .236 | .210 | .089 | .202 | .243 | .191 |
| | Avg | .698 | **.701** | .650 | .606 | .627 | .487 | .554 | .555 | .523 |
| 00% | FM | .861 | **.864** | .861 | .761 | .801 | .727 | .842 | .760 | .751 |
| | CA | .870 | **.879** | .870 | .820 | .865 | .856 | .863 | .862 | .860 |
| | SS | .364 | .357 | **.364** | .236 | .217 | .189 | .307 | .266 | .258 |
| | Avg | .698 | **.700** | .698 | .606 | .628 | .591 | .671 | .629 | .623 |

Note: The numbers are average accuracy over three runs.

## 7 CONCLUSION AND FUTURE WORK

This paper shows two methods for improving the adversarial robustness of federated learning under a majority adversary regime. Empirical results in various settings and against a broad class of attacks demonstrate the proposed methods' effectiveness. Additional theoretical analysis is conducted under the Mimic-Shift attack regime, showing conditions under which the proposed method helps. Further exploring the limitations of learning with majority adversaries is a good next step.

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
