# OpenReview forum: "Robust Federated Learning with Majority Adversaries via Projection-based Re-weighting"
_ICLR.cc/2023/Conference — Submitted to ICLR 2023_

### Official Review · Reviewer_ZCpp · 2022-10-24

**Confidence:** 2
**Correctness:** 3
**Technical Novelty And Significance:** 3
**Empirical Novelty And Significance:** 2
**Recommendation:** 5

**Clarity, Quality, Novelty And Reproducibility:**

Clarity: Some clarification about the unique problem setting and empirical results can be further added or improved.

Quality: The technical quality of this work is of high quality with solid theoretical results.

Novelty: This work proposed a two-staged defense using filtering and re-weighting to tackle the majority adversary setting. The conceptual novelty of considering the majority adversary setting is high. However, the current presentation does not highlight the technical novelty of the proposed method, as filtering and reweighting seem to be some common strategies used in previous literature about defending.

Reproducibility: The current version of this work does not provide its source code. With limited experimental details, it is hard to adjust the reproducibility of the proposed method. It is encouraged to open-source the code or provide more implementation setups than hyperparameter sets.

**Strength And Weaknesses:**

Strength:
1. This paper explores a new setting, which differs from the assumption in most previous literature, i.e., the majority adversary setting. It is practical as indicated in Kairouz et al. (2021) [1].
2. The proposed two-staged defense shows empirical effectiveness on several baseline methods.

Weaknesses:
1. The motivation for the proposed second-stage reweighting is not very clear. It could be better improved by adding a more intuitive example or illustration that the reweighting mechanism is the target choice for defending the mimic-shift attack.
2. The technical novelty of the current presentation is limited, as the reweighting mechanism or filtering is commonly used to defend against poisoning attacks. It is encouraged to add more discussion about the rationality of adopting reweighting in this problem.
3. It may lack some empirical evidence to verify the illustration in Figure 2.

Other questions/comments:
1. Is there any empirical evidence supporting "the decentralized nature means that it is relatively straightforward for the adversary to be the majority and thus break existing defense"?
2. It is better to link some major clarification with that empirical evidence or support with reference.
3. Why the performance of aggregators can still outperform the oracle when the adversary is 0% in Table 2?

[1] Peter Kairouz, H Brendan McMahan, Brendan Avent, Aurelien Bellet, Mehdi Bennis, Arjun Nitin Bhagoji, Kallista Bonawitz, Zachary Charles, Graham Cormode, Rachel Cummings, et al. Advances and open problems in federated learning. Foundations and Trends® in Machine Learning, 14(1–2):1–210, 2021.

**Summary Of The Paper:**

This paper focuses on robust federated learning against adversarial clients, which hinder the convergence and performance of federated learning via injecting malicious local updates. Unlike other previous works, this paper considers the majority adversarial setting, where the adversarial clients are more than those benign clients and thus break the existing defenses. Given the assumption of having few trusted clients for reference, this work proposes a two-staged defense combining filtering and re-weighting to defend against a broad class of attacks. Extensive experiments are conducted to verify the effectiveness of the proposed method, and theoretical analyses are provided.

**Summary Of The Review:**

Overall, this paper proposes an effective method for the majority adversary setting of the robust federated learning problem. This paper could be further improved to make the clarification clearer and stronger, as well as highlight the technical novelty of the proposed method by adding more conceptual and intuitive comparisons.

---

> ### Author Response · Authors · 2022-11-17
> **Response to Reviewer ZCpp**
>
> Thanks for the comments. We have clarified the novelty and the contribution of our work in the response. We have also added the suggest empirical evaluation of the two cases in Figure 2 to the post-rebuttal version.
>
> &nbsp;
>
> **The motivation for the proposed second-stage reweighting is not very clear. It could be better improved by adding a more intuitive example or illustration that the reweighting mechanism is the target choice for defending the mimic-shift attack.**
>
> Selectively down-weighting potential malicious clients can mitigate their impacts. We would like to remind the reviewer that our main contribution in the second stage is on “how” to select clients to down-weight.
>
> &nbsp;
>
> **The technical novelty of the current presentation is limited, as the reweighting mechanism or filtering is commonly used to defend against poisoning attacks. It is encouraged to add more discussion about the rationality of adopting reweighting in this problem.**
>
> Although weighting mechanisms have been considered in many previous works, our idea of down-weighting clients that have high influence on the aggregation result is novel and has not yet been explored.
>
> &nbsp;
>
> **It may lack some empirical evidence to verify the illustration in Figure 2.**
>
> Thanks for the suggestion. We have added additional experiments to explore the two cases in Figure 2 to Appendix C.16 of the post-rebuttal version. We find that the $2\theta^* \leq \theta$ case in Figure 2(a) happens with 30\% - 80\% adversaries, and the general case in Figure 2(b) happens with 10\% - 20\% adversaries.
>
> &nbsp;
>
> **Is there any empirical evidence supporting "the decentralized nature means that it is relatively straightforward for the adversary to be the majority and thus break existing defense"? It is better to link some major clarification with that empirical evidence or support with reference.**
>
> Section 5.1 in [1] suggests that unreliable devices might introduce new attack surfaces at training time in federated learning, compared to distributed data center learning and centralized learning schemes.
>
> &nbsp;
>
> **Why the performance of aggregators can still outperform the oracle when the adversary is 0% in Table 2?**
>
> In section 6.3, we hypothesize that the improved results can come from the up-weighting of the underrepresented clients whose influence score is small. E.g. Li et al., 2020 use a weighting strategy to improve fairness and observe similar results.
>
> &nbsp;
>
> **The current version of this work does not provide its source code. With limited experimental details, it is hard to adjust the reproducibility of the proposed method. It is encouraged to open-source the code or provide more implementation setups than hyperparameter sets.**
>
> Thanks for the suggestion. We have linked our source code anonymously in the private comment to reviewers. We will clean up and open-source our code upon publication.
>
> &nbsp;
>
> **Reference**
>
> [1] Kairouz, Peter, et al. "Advances and open problems in federated learning." Foundations and Trends® in Machine Learning 14.1–2 (2021): 1-210.

---

### Official Review · Reviewer_pinJ · 2022-10-25

**Confidence:** 3
**Correctness:** 3
**Technical Novelty And Significance:** 3
**Empirical Novelty And Significance:** 3
**Recommendation:** 6

**Clarity, Quality, Novelty And Reproducibility:**

**Main argument**

For the problem the paper is trying to solve, I cast doubts on whether the majority adversary is universal in FL systems, which may limit the performance of methods designed for fighting against it in real-case scenarios. As the author claims in the introduction, the client availability issue may limit the number of clients participating in a round. However, majority adversaries will still happen with a small probability because the server randomly selects clients, for which it is hard for adversaries to be majority even in a round when the whole system is under minority adversaries setting.

The idea of measuring good/bad clients with the projection of average gradient on them is innovative; however, I'm not sure why it works even in minority adversary setting where the method using projection should misjudge benign users as an adversary with more detriments on the resulting average gradient.

The theory misses the part proving that Mimic-shift will damage the accuracy of federated learning under the majority adversary setting, which I think is not fully confirmed with experiments as well because there are not enough different hyper-parameter studies for Mimic-Shift and Mimic-Shift-Par.

From the experiment, MIMIC-Shift-Par is an effective and practical attack method. However, there is no explanation for why it can attack effectively without knowing the benign users. Randomly eavesdropping 20% of clients per round when the majority of clients are adversary means the clients being eavesdropped on are also mainly adversaries, which leads to great bias when assuming the averaged gradient of eavesdropped clients represents the average gradient of benign users. In that situation, a theory explaining why MIMIC-Shift-Par works is welcomed.

For the experiments, the following should be addressed.

1. evaluate the damage Mimic-shift can cause when a relatively small proportion of adversaries exist in Federated learning systems can cause (or analysis/measurement of the probability majority of FL systems will be adversaries with random sampling clients)?
2. It is confusing whether "Oracle" means no attacks for all clients or not. If so, in Table 6 the accuracy of "Oracle" is 0.870, different from 0.869 in Table 1; what causes that difference in the same setting?
3. explain why the proposed method also does well in the minority adversary setting.
4. Ablation study of phase 1 and phase 2 demonstrating that filtering and re-weight both play vital roles in defending against attacks.
5. why the variance of all methods in Table 1 is 0.001, is the effect of attacks really so robust to randomness as observed from the variance?
6. results in Mimic-Shift and Mimic-Shift-Par with different hyper-parameters will help to confirm the effectiveness of these two attacks.

Minor comments:

1. Page 9: In 6.3 DEFENSE AGAINST MIMIC-SHIFT ATTACK, "Our method occationally outperforms the oracle." Does "occationally" here mean "occasionally"?
2. In Table 1, I think the accuracies of standard attacks are suitable to be put here instead of in the appendix.
3. In Table 2, I think 40% adversary is a little too high for minority adversary setting, maybe 20% adversary is a better choice.

**Strength And Weaknesses:**

**Strength and Weaknesses:**

This paper does well in: (1) clearly explain the intuition behind their proposed methods. (2) the idea of judging benign/adversarial clients with projection on average gradient is innovative. (3) the proposed attack and defend method seem promising under majority adversary setting observed from the results of experiments.

As for the weaknesses, I have several concerns requiring the authors to clarify: (1) the paper doesn't convincingly clarify the extensiveness of the setting (Majority adversaries can be a common situation in Federated learning systems). (2) some parts of the results of experiments are not clearly analyzed in the paper. (3) The theory does not provide a theoretic guarantee for dropping accuracy caused by Mimic-shift. (4) the effect of filtering is not demonstrated in practice and missing of that may makes the validity of re-weight phase less convincing.

**Summary Of The Paper:**

This paper proposes a new type of attack named Mimic-Shift and designs a two-phase defend framework under majority adversary setting. The proposed framework combines classic filtering method defending standard attack together with a new re-weighting method reducing the influence of Mimic-Shift . In order to weaken the effect of averaged gradients from adversaries while strengthening the effect of average gradient from benign users, the proposed method utilizes projection to judge whether a client is a benign user or an adversary client in the setting where exists majority adversaries. The effectiveness of Mimic-shift and Mimic-shift-Par is shown and compared with other kinds of attacks. And the performance of the proposed method against attacks is demonstrated with the report of accuracies in several classic benchmarks of Federated learning (FEMNIST, CelebA, and Shakespeare), compared with some baselines designed for defending against standard attacks.

**Summary Of The Review:**

Overall, this paper proposes a practically effective way of attacking and a method against it; however, there is some flaws in both theory and experiments. Given these clarifications in an author response and some necessary complementary experiments, I would be willing to increase the score.

---

> ### Author Response · Authors · 2022-11-17
> **Response to Reviewer pinJ (2/2)**
>
> **Why the variance of all methods in Table 1 is 0.001, is the effect of attacks really so robust to randomness as observed from the variance?**
>
> Yes, the small variance is consistent with the previous work [2] on Mimic attacks on FedAvg aggregators (Table 2, row 1).
>
> &nbsp;
>
> **Does "occationally" here mean "occasionally"?**
>
> Yes, we will fix the typo. Thank you.
>
> &nbsp;
>
> **In Table 1, I think the accuracies of standard attacks are suitable to be put here instead of in the appendix.**
>
> Thanks for the suggestion. We will find space in the main paper.
>
> &nbsp;
>
> **In Table 2, I think 40% adversary is a little too high for minority adversary setting, maybe 20% adversary is a better choice.**
>
> Thanks for the suggestion. We would like to remind the reviewer that our work focuses on majority adversary settings, where existing defenses can fail. Standard methods [2] may be more appropriate if designers expect minority adversaries.
>
> We have added additional experimental results with 20% adversaries to Appendix C.6. Our results show that our defense loses no accuracy on FEMNIST and CelebA datasets. Our defense does not apply to the Shakespeare dataset with 20% adversaries, possibly because benign gradients in the Shakespeare experiment are not scattered enough.
>
> &nbsp;
>
> **Reference**
>
> [1] Bonguet, Adrien, and Martine Bellaiche. "A survey of denial-of-service and distributed denial of service attacks and defenses in cloud computing." Future Internet 9.3 (2017): 43.
>
> [2] Karimireddy, Sai Praneeth et al. “Byzantine-Robust Learning on Heterogeneous Datasets via Bucketing.” ICLR (2022).

---

> ### Author Response · Authors · 2022-11-17
> **Response to Reviewer pinJ (1/2)**
>
> Thanks for the comments. We have clarified our majority adversary setting and the behavior of our defense under a minority adversary setting in the response. We have also added empirical&theorical verifications of our Mimic-Shift attack, additional ablation studies, and experiments with 20% adversaries
>
> &nbsp;
>
> **I cast doubts on whether the majority adversary is universal in FL systems. As the author claims in the introduction, the client availability issue may limit the number of clients participating in a round. However, majority adversaries will still happen with a small probability because the server randomly selects clients, for which it is hard for adversaries to be majority even in a round when the whole system is under minority adversaries setting. Experiments shall evaluate the damage Mimic-shift can cause when a relatively small proportion of adversaries exist in Federated learning systems can cause (or analysis/measurement of the probability majority of FL systems will be adversaries with random sampling clients)?**
>
> Our setting is different from the minority adversaries setting mentioned by the reviewer. Majority adversaries can exploit the client availability issue and directly connect fake clients to federated learning systems to outnumber benign clients in the sampling pool that is available to servers. In this case, a majority of clients in the sampling pool are malicious, so the majority of randomly sampled clients from the pool can be malicious with a high probability.
>
> Here, fake clients can be distributed networked devices whose number is large, as is considered in common distributed denial-of-service attacks (Section 1). Distributed denial-of-service attacks can launch massive distributed networked devices to overwhelm servers [1].
>
> &nbsp;
>
> **The idea of measuring good/bad clients with the projection of average gradient on them is innovative; however, I'm not sure why it works even in minority adversary setting.**
>
> We discussed the conditions under which our approach works under a minority setting in Section 5.2.1. We agree with the reviewer that the projection operation could misjudge benign users as an adversary under a minority setting, but our approach may not break with 50% adversaries. This is because benign updates can be scattered and far away from the average, decreasing the projection scores of benign updates. As long as benign updates are sufficiently scattered and receive lower project scores than malicious updates (e.g., from a 40% adversary), our approach remains effective.
>
> &nbsp;
>
> **The theory misses the part proving that Mimic-shift will damage the accuracy of federated learning under the majority adversary setting, which I think is not fully confirmed with experiments as well because there are not enough different hyper-parameter studies for Mimic-Shift and Mimic-Shift-Par.**
>
> The intuition behind Mimic-Shift and Mimic-Shift-Par is the same. We mirror a less biased average update from all sampled clients using a more biased average update from reference clients or the 20% eavesdropped clients, making the mirroring result even more biased. Here, the more biased an update is, the further away it is from the true expectation. Further, Table 1 provides experiments showing that Mimic-shift will damage the accuracy of federated learning under the majority adversary setting (80%).
>
> We have completed additional hyper-parameter studies to show that Mimic-Shift attacks are effective with different numbers of reference clients per round (Appendix C.15 of the post-rebuttal version), decreasing the accuracies from 0.861 to 0.621 (2 reference clients) - 0.685 (12 reference clients). We have also added a Theorem to show the necessity of using fewer reference clients in Mimic-Shift attacks (Theorem 13, Appendix C.15 of the post-rebuttal version).
>
> &nbsp;
>
> **It is confusing whether "Oracle" means no attacks for all clients or not. If so, in Table 6 the accuracy of "Oracle" is 0.870, different from 0.869 in Table 1; what causes that difference in the same setting?**
>
> In section 6.3, we hypothesize that the improved results can come from the up-weighting of the underrepresented clients whose influence score is small. E.g. Li et al., 2020 use a weighting strategy to improve fairness and observe similar results.
>
> &nbsp;
>
> **Ablation study of phase 1 and phase 2 demonstrating that filtering and re-weight both play vital roles in defending against attacks.**
>
> Thanks for the suggestion. We have completed additional ablation studies on the FEMNIST. Appendix C.7 of the post-rebuttal version shows that removing phase 2 decreases the accuracy by 0.219 under Mimic-Shift attacks, and removing phase 1 decreases accuracies by up to 0.860 under Gaussian and Sign-flipping attacks.

---

> > ### Comment · Reviewer_pinJ · 2022-12-12
> > **Confirming response**
> >
> > Thanks for the clarification and revisions and sorry for the late response:
> >
> > 1. I'm sorry that I misunderstand that 40% is for showing a less majority client but still under majority adversaries setting. I thought 40% is for showing your defend is available even under minority adversary setting.
> > 2. The experiments and explanation together alleviate my concern about the relatively minority adversary setting. Thank you for it.
> > 3. The additional hyper-parameter studies convince me for the robustness of those two kinds of attacks (Mimic-Shift and Mimic-Shift-Par). Thank you for your revision.
> > 4. I still think the term "Oracle" is in need of more clarification in your paper.
> > 5. Thank you for taking my advise to add the ablation studies.
> > With the revision and explanation, I am willing to raise the recommendation to 6.

---

> > > ### Author Response · Authors · 2022-12-12
> > > **Response to Reviewer pinJ**
> > >
> > > Thanks for the reply and the suggestion. We will (1) clarify the setting of "Oracle" and (2) add experimental verifications of our hypothesis regarding the accuracy of "Oracle".
> > >
> > > We would very much appreciate it if the reviewer could update the recommendation in the initial review.

---

### Official Review · Reviewer_odUz · 2022-10-27

**Confidence:** 4
**Correctness:** 3
**Technical Novelty And Significance:** 2
**Empirical Novelty And Significance:** 2
**Recommendation:** 6

**Clarity, Quality, Novelty And Reproducibility:**

The paper will be stronger if more baseline attacks and adaptive attacks are evaluated.

**Strength And Weaknesses:**

Strength

+ A new robust FL method is proposed.

+ Theoretical analysis is performed on the proposed method to show its robustness.

Weakness

- Some baseline attacks are not evaluated, e.g., [A]

- Adaptive attacks are not considered. Defending against existing attacks is not hard. But the challenge is that attackers are always adaptive to the defenses. So we should evaluate adaptive attacks that adapt to the defense.

- FLTrust description seems not correct. I thought the FLTrust does not only consider cosine similarity. It also uses ReLU to clip cosine similarity.

[A] local model poisoning attacks to byzantine-robust federated learning. In USENIX Security Symposium, 2020.

**Summary Of The Paper:**

The paper proposes a new method to build robust FL. The paper performs theoretical analysis and evaluation on multiple datasets and baselines.

**Summary Of The Review:**

The paper proposes a new robust FL, performs theoretical analysis on the method, and compares with some baselines. The paper would be stronger if more baseline attacks and adaptive attacks are evaluated.

---

> ### Author Response · Authors · 2022-11-17
> **Response to Reviewer odUz**
>
> Thanks for the comments. We have clarified adaptive attacks, included the suggested experiments in the response, and corrected the FLTrust description.
>
> &nbsp;
>
> **Some baseline attacks are not evaluated.**
>
> Thanks for the suggestion. We have included additional experiments in the post-rebuttal version (Appendix C.5) with local model poisoning attacks, showing that our approach remains effective. Our results show that our defense only loses by up to 2.7% accuracy against 80% adversaries, compared to the oracle aggregator that only accept benign updates.
>
> &nbsp;
>
> **Adaptive attacks are not considered.**
>
> Our experiments included the Mimic-Shift-Var attack, which is particularly designed as a strong adaptive attack for our proposed defense. The Mimic-Shift-Var attack adds variance to malicious updates via client-wise mirroring. Such variance reduces the influence of each malicious update on aggregation results. We showed that our approaches remain effective in Appendix C.3 of the pre-rebuttal version.
>
> &nbsp;
>
> **FLTrust description seems not correct.**
>
> Thanks for the suggestion. We have corrected the description in the post-rebuttal version.

---

### Official Review · Reviewer_VsGK · 2022-10-30

**Confidence:** 4
**Correctness:** 2
**Technical Novelty And Significance:** 2
**Empirical Novelty And Significance:** 3
**Recommendation:** 3

**Clarity, Quality, Novelty And Reproducibility:**

As mentioned above, the writing is not great: it ignores provably approaches to resist attacks against general poisoning attacks (that cover attacks in federated learning as a special case). The theory part is also not easy to read and interpret/ understand.

You say in the introduction that you need secure hardware. Where exactly do you use secure hardware? Are you only referring to the trusted parties? That is not what other papers mean by secure hardware (enclave) usually.

Some comments:

Why use F for the loss function, and not, say, L?

You say “The Mimic-Shift attack is also difficult to detect because the malicious g′ has the same distance to the reference ¯g_R as the benign ¯g.”
But this is an extremely simple way of detecting things and says nothing about a detector that does something slightly more intelligent.

I strongly recommend rewriting the theory section and explaining/discussing/interpreting your assumptions as you make them. Also, the theorem statements are not clear. With so much math notation and words hanging there, it is hard to keep track of what the theorems are really proving. There are also specific assumptions in the theorem statements that need to be discussed and  explained why making them is reasonable.


**Strength And Weaknesses:**

Strength: aiming to provide solutions for federated learning that resist many corruptions. It is also interesting to combine two different methods as a defense, as they could work well together.

Weakness: the evaluation is not against adaptive attacks (who know the defense). The theoretical part has too many assumptions that are hard to interpret.


**Summary Of The Paper:**

The paper studies attacks and defenses for federated learning. In particular,
1) The paper first observes that a “filtering based” attack that uses a few “trusted clients” can help bypass some attacks.
2) Then it presents an attack called “mimic shift” that somehow improves previous attacks known as “mimic” attacks and bypasses the simple defense.
3) Then it improves the defense to add a re-weighting step and shows that it can defend against some standard attacks. The idea is to use a bunch of trusted nodes and use them to approximate the statistics of the gradient, and use that to filter out the outliers. Also, re-weight the nodes to balance out their influence on the gradients.
The paper does some theoretical analysis of their defense (under quite a few assumptions). It also does some experimental study for the 3rd part above.

I have provided more details below, but I think in summary, the paper suffers from an issue that many papers in adversarial/robust learning do: it does not study “adaptive” attacks that are specifically designed with the knowledge of a defense. That is why these works usually fall into a cat and mouse game of defense/attack/defense/etc. By now there are provably approaches in robust (poisoning resistant) learning that apply to federated learning as well. For example look up [1] and other papers on “robust statistics”. This paper ignores that line of work and merely studies heuristic approaches with much weaker guarantees

[1] Jia, Jinyuan, Xiaoyu Cao, and Neil Zhenqiang Gong. "Intrinsic certified robustness of bagging against data poisoning attacks." AAAI’21

In fact, the paper explicitly says “Other attacks, including data poisoning …  and backdoor (..),
are beyond the scope of this work.”
But why is that so? Isn't their framing more general than yours?


**Summary Of The Review:**

In summary, I think the main question of the paper is interesting. But the methods are not fundamentally new, are not adding real guarantees (as they can be broken in the next paper) even though such provable approaches exist, and the paper’s writing in the theory section is not great. Therefore, I cannot support it for ICLR acceptance, even though I acknowledge the merits of the paper.

---

> ### Author Response · Authors · 2022-11-17
> **Response to Reviewer VsGK (2/2)**
>
> **You say “The Mimic-Shift attack is also difficult to detect because the malicious g′ has the same distance to the reference ¯g_R as the benign ¯g.” But this is an extremely simple way of detecting things and says nothing about a detector that does something slightly more intelligent.**
>
> We assume the reviewer refers to a defense that detects duplicated updates. Adding a small Gaussian noise to each malicious update can circumvent such a strategy. We conducted additional experiments to show that the Mimic-shift attack and our defense both remain effective. The noise mean is 0, and the noise variance is 0.01 times the update magnitude.
>
> | Aggregator | FEMIST | CelebA | Shapespeare |
> | ----------- | ----------- | ----------- | ----------- |
> | FedAvg | .620 | .797 | .169 |
> | Ours | .840 | .876 | .360 |
>
> &nbsp;
>
> **But the methods are not fundamentally new.**
>
> Although weighting mechanisms have been considered in many previous works, our idea of down-weighting clients that have high influence on the aggregation result is novel and has not yet been explored.
>
> &nbsp;
>
> **The methods are not adding real guarantees as they can be broken in the next paper.**
>
> We have included extensive empirical study with five model poisoning attacks, including a strong adaptive Mimic-Shift-Var attack that is particularly designed for our defense. All the empirical evaluations suggest that our defense is hard to break under a majority adversary setting.
>
> &nbsp;
>
> **Reference**
>
> [1] Xie, Chulin, et al. "Crfl: Certifiably robust federated learning against backdoor attacks." International Conference on Machine Learning. PMLR, 2021.
>
> [2] Kairouz, Peter, et al. "Advances and open problems in federated learning." Foundations and Trends® in Machine Learning 14.1–2 (2021): 1-210.
>
> [3] Karimireddy, Sai Praneeth et al. “Byzantine-Robust Learning on Heterogeneous Datasets via Bucketing.” ICLR (2022).
>
> [4] Wu, Jingfeng, et al. "On the noisy gradient descent that generalizes as sgd." International Conference on Machine Learning. PMLR, 2020.
>
> [5] Xie, Cong, Sanmi Koyejo, and Indranil Gupta. "Zeno: Distributed stochastic gradient descent with suspicion-based fault-tolerance." International Conference on Machine Learning. PMLR, 2019.
>
> [6] Xie, Cong, et al. "Cser: Communication-efficient sgd with error reset." Advances in Neural Information Processing Systems 33 (2020): 12593-12603.

---

> > ### Comment · Reviewer_VsGK · 2022-11-27
> > **Confirming response**
> >
> > Thanks for the clarification's. Some responses back:
> >
> > - Regarding adaptivity: what i mean is that you need to define your defense and allow the adversary to know your defense and still show that the defense works. you first show a type of defense, and then show an attack on it, and then another dense against your attack. this game can continue further if adaptive attacks are not prevented using defenses that allow attackers to know them.
> >
> > - you say your problem setting is fundamentally different from from the AAA21 paper. I respectfully (strongly) disagree. Their setting seems more general to me as a defense. Namely, they allow a number of parties to submit completely malicious data (or models that are completely malicious). OF course they might not be able to defend as many corrupted parties as you could, but that is because they allow arbitrary attacks.
> >
> > The part I quoted "The Mimic-Shift attack is also difficult to detect" was to give an example that the word "difficult" is used without mathematical formalism. Do you mean computationally hard even using any polynomial time algorithm? I don't think that is what you mean.
> >
> > My comments regarding the presentation in the theory section still stand.

---

> > > ### Author Response · Authors · 2022-11-28
> > > **Response to Reviewer VsGK (2/2)**
> > >
> > > **The presentation in the theory section is unclear. The assumptions and theories are not explained.**
> > >
> > > We have discussed Assumption 1 in the paper. We further add more discussions on Assumptions 2-5, which are common in existing works.
> > >
> > > In Assumption 1, we assume all users have the same number of samples because the difference in sample size can be merged into the difference between gradients.
> > >
> > > In Assumption 2, we assume that the estimated gradient on each client comes from a Gaussian distribution with a truncation threshold. The Gaussian gradient assumption is common [1]. We assume truncation thresholds to make Assumption 2 consistent with Assumption 5, which assumes bounded gradient norms. Then, we further assume that the mean and variance of the Gaussian distribution on each client come from distributions with finite expected values and finite variance.
> > >
> > > Assumptions 3-5 assume that the gradient variances and norms are bounded, which is common [2, 3].
> > >
> > > Our theoretical analysis is composed of three parts. We start with the false-positive filtering issue in State 1 (i.e., benign updates are removed). Then, we proceed to the re-weighting phase, showing conditions under which malicious updates get the highest scalar projections and are, therefore, down-weighted most. We further combine the results on false-positive filtering and re-weighting and show the convergence results of our approach. We briefly re-state the discussion in Section 5:
> > >
> > > ***False positive filtering (Section 5.1)***: Lemma 6 suggests that the probability of filtering out a benign gradient decreases at a rate of $\mathcal{O}(\frac{1}{c^2})$, where $c$ controls the filtering radius, assuming a condition $c \geq 2 \cdot  \frac{\|\mathbb{E}[\mu] - \bar{\mu}\|}{s_R}$ holds. Then, Lemma 7 further shows the probability of violating the assumed condition $c \geq 2 \cdot  \frac{\|\mathbb{E}[\mu] - \bar{\mu}\|}{s_R}$ decreases at a rate of $\mathcal{O}(\frac{1}{N_R^2})$, where $N_R$ is the number of reference clients. Theorem 9 combines Lemma 7 and 8,  suggesting that the risk of filtering out a benign gradient decreases quickly w.r.t. $c$ and $N_R$, enabling our strategy of using a small number of reference clients and a conservative $c$.
> > >
> > > ***Down-weighting malicious updates (Section 5.2)***: Theorem 10 outlines conditions under which the probability of down-weighting malicious gradients more than benign gradients increases as the adversary owns more clients and samples in a system. Such a property suggests that the re-weighting phase is effective against majority adversaries.
> > >
> > > ***Convergence result (Section 5.3)***: Theorem 12 shows that our method converges to a neighborhood around the optimum under a convex setting, whose size shrinks with fewer false-positive filtering (Theorem 9) and more down-weighting (tuning $k$ and $τ$ in Algorithm 2) with higher probability (Theorem 10) on the malicious gradients. Converging to a neighborhood is common in previous noisy gradient descent studies [4, 5].
> > >
> > > &nbsp;
> > >
> > > **Reference**
> > >
> > > [1] Wu, Jingfeng, et al. "On the noisy gradient descent that generalizes as sgd." International Conference on Machine Learning. PMLR, 2020.
> > >
> > > [2] Xie, Cong, Sanmi Koyejo, and Indranil Gupta. "Zeno: Distributed stochastic gradient descent with suspicion-based fault-tolerance." International Conference on Machine Learning. PMLR, 2019.
> > >
> > > [3] Xie, Cong, et al. "Cser: Communication-efficient sgd with error reset." Advances in Neural Information Processing Systems 33 (2020): 12593-12603.
> > >
> > > [4] Wang, Yunjuan, et al. “Robust learning for data poisoning attacks.” International Conference on Machine Learning. PMLR, 2021.
> > >
> > > [5] He, Lie, et al. “Byzantine-robust decentralized learning via self-centered clipping.” ArXiv, abs/2202.01545, 2022.

---

> > > ### Author Response · Authors · 2022-11-28
> > > **Response to Reviewer VsGK (1/2)**
> > >
> > > Thanks for replying. We have clarified the adaptive attack and discussed the difference between our work and the AAAI21 paper. We have also added more details about the Mimic-Shift attack and our theoretical analysis.
> > >
> > > &nbsp;
> > >
> > > **Adaptivity is misinterpreted.**
> > >
> > > Our adaptive attack (i.e., Mimic-Shift-Var) follows the adaptivity definition mentioned by the reviewer. Adaptive adversaries know our defense and aim to (1) circumvent the filtering phase and (2) reduce the projection scores of malicious updates in the re-weighting phase. Reduced projection scores allow malicious updates to be down-weighted less.
> > >
> > > The adaptive Mimic-Shift-Var (Section 6.1) attacks achieve the two aforementioned goals simultaneously by mirroring benign updates w.r.t. the aggregated reference updates in a client-wise manner because (1) mirroring results still stay within the filtering circle in stage 1 and (2) client-wise mirroring diversifies malicious updates and reduces the projections scores. Note that the aggregated malicious updates from Mimic-Shift-Var attacks are supposed to be the same as those of Mimic-Shift attacks.
> > >
> > > We showed that our approach remains effective in Appendix C.3 of the pre-rebuttal version.
> > >
> > > &nbsp;
> > >
> > > **The setting in the AAA21 paper seems more general to me as a defense.**
> > >
> > > The difference between our setting and the setting in the AAAI21 paper is two-fold, and neither works are more general than the other one. The AAAI21 paper assumes centralized learning, but our work focuses on federated learning. In addition, we consider model poisoning attacks, while the AAAI21 paper considers data poisoning attacks. The two major differences make applying the AAAI21 paper to our setting difficult.
> > >
> > > First, global bagging is infeasible in federated learning. If we let each client trains a model as a base classifier, clients need to submit their data samples to other clients or servers to get majority voting. However, submitting data samples is prohibitive in federated learning due to privacy concerns.
> > >
> > > On the other hand, local bagging is not effective in our setting. Local bagging means each client trains an ensemble model locally and servers use federated averaging to aggregate the ensemble models from clients. However, under our threat model that considers model poisoning attacks, adversaries could directly corrupt all base classifiers in the ensemble and subsequently break voting results. Our empirical results show that the local bagging strategy is not effective. The metric is predictive accuracy, which is considered in our work and is a special case of the certification accuracy in the AAAI21 paper with certification radius $r$ = 0.
> > >
> > > | Aggregator | FEMIST | CelebA | Shapespeare |
> > > | ----------- | ----------- | ----------- | ----------- |
> > > | Local Bagging | .607 | .795 | .152 |
> > > | Ours | .840 | .877 | .360 |
> > >
> > > &nbsp;
> > >
> > > **The method in the AAAI21 paper might not be able to defend as many corrupted parties as you could, but that is because they allow arbitrary attacks.**
> > >
> > > We would like to remind the reviewer that our work considers a different threat model (i.e., model poisoning attacks from majority adversaries), and the AAAI21 paper is not directly applicable as is discussed above. Our threat model focuses on majority adversaries that can perform Mimic-type attacks. None of the existing works considers this threat model or can defend against such adversaries. In addition, we conducted extensive empirical evaluations of our defense with a total of five attacks, including an adaptive attack.
> > >
> > > &nbsp;
> > >
> > > **Difficulties in detecting Mimic-Shift attacks.**
> > >
> > > That’s a good point. We further clarify our claim: Mimic-Shift attacks are difficult to detect via similarity-based methods using Euclidean distance or cosine similarity because the malicious updates are close to the reference updates. Our empirical results show that Mimic-Shift attacks can circumvent the distance-based filtering phase and the cosine-similarity-based FLTrust.

---

> ### Author Response · Authors · 2022-11-17
> **Response to Reviewer VsGK (1/2)**
>
> Thanks for the comments. We have clarified adaptive attacks, discussed the difference between our approach and certified defenses, and added more details about our method in the response.
>
> &nbsp;
>
> **It does not study “adaptive” attacks that are specifically designed with the knowledge of a defense. The evaluation is not against adaptive attacks (who know the defense).**
>
> Our experiments included the Mimic-Shift-Var attack, which is particularly designed as a strong adaptive attack for our proposed defense. The Mimic-Shift-Var attack adds variance to malicious updates via client-wise mirroring. Such variances reduce the influence of each malicious update on aggregation results. We showed that our approaches remain effective in Appendix C.3 of the pre-rebuttal version.
>
> &nbsp;
>
> **This paper ignores that line of work on certified robustness and merely studies heuristic approaches with much weaker guarantees. It ignores provably approaches to resist attacks against general poisoning attacks (that cover attacks in federated learning as a special case).**
>
> Thanks for sharing the AAAI’21 paper. However, our settings are fundamentally different. Certifiable federated learning [1] provides sample-wise robustness guarantees during test time. In contrast, our work aims to protect population-wise utility at training time.
>
> &nbsp;
>
> **In fact, the paper explicitly says “Other attacks, including data poisoning … and backdoor (..), are beyond the scope of this work.” But why is that so? Isn't their framing more general than yours?**
>
> Model poisoning attacks, data poisoning attacks, and backdoor attacks are distinct types of adversarial attacks and are considered non-overlapping, i.e., none of them are more general than the other two[2]. We focus on model poisoning attacks as one of the important settings and consider how the recent Mimic-type attacks from this class can circumvent existing defenses [3].
>
> &nbsp;
>
> **The theoretical part has too many assumptions that are hard to interpret. There are also specific assumptions in the theorem statements that need to be discussed and explained why making them is reasonable**
>
> Assumptions 2-5 are common in the machine learning and federated learning literature [4, 5, 6]. We explained the first assumption: To simplify tedious notation, we assume all users have the same number of samples because the difference in sample size can be merged into the difference between gradients.
>
> We don't make additional assumptions in the theorems. The conditions in the theorems are discussed in the text that follows the theorems.
>
> &nbsp;
>
> **The theory part is also not easy to read and interpret/ understand. I strongly recommend rewriting the theory section and explaining/discussing/interpreting your assumptions as you make them. Also, the theorem statements are not clear. With so much math notation and words hanging there, it is hard to keep track of what the theorems are really proving.**
>
> In the pre-rebuttal version, we summarized each theoretical result following the theorems and lemmas and briefly discussed the high-level proof strategy. We also provide figures to ease the reading. We are happy further improve the readability.
>
> &nbsp;
>
> **Where exactly do you use secure hardware? Are you only referring to the trusted parties? That is not what other papers mean by secure hardware (enclave) usually.**
>
> We assume reference clients adopt secure hardware (e.g., enclave) on their devices (e.g., mobile phones). Section 1 includes references to secure hardware on commercial mobile phones (e.g., Pixel and iPhone). We do not consider trusted parties.
>
> Thus, we assume that this secure hardware guarantees trusted gradient computation for the reference clients, while other clients' gradient computations may not be trusted. Servers can know which clients estimate their gradients inside secure hardware via remote attestation (https://www.intel.com/content/www/us/en/developer/tools/software-guard-extensions/attestation-services.html).
>
> &nbsp;
>
> **Why use F for the loss function, and not, say, L?**
>
> Thanks for the suggestion. We will use $\ell$ for the loss function.

---

### Author Response · Authors · 2022-11-17
**Revision Summary**

We thank all the reviewers for their helpful comments and suggestions. We have revised our submission accordingly. The revised version includes the following:

- Empirical evaluations against local model poisoning attacks (Appendix C.5).

- Empirical evaluations under a 20% adversary setting (Appendix C.6).

- Ablation studies of the two phases of our defense (Appendix C.7).

- Empirical evaluations of Mimic-Shift with different hyper-parameters, which is the number of reference clients (Appendix C.15).

- A theoretical analysis explains why it is necessary to use fewer reference clients in our Mimic-Shift attack (Appendix C.15).

- Empirical evaluations of gradient angles in Figure 2 (Appendix C.16).

---

### Decision · Program_Chairs · 2023-01-20

**Decision:**

Reject

**Justification For Why Not Higher Score:**

According to my expertise and reviewing process, this paper should belong to a Reject.

**Justification For Why Not Lower Score:**

According to my expertise and reviewing process, this paper should belong to a Reject.

**Metareview: Summary, Strengths And Weaknesses:**

This paper proposes a new type of attack named Mimic-Shift and designs a two-phase defend framework under majority adversary setting. The proposed framework combines classic filtering method defending standard attack together with a new re-weighting method reducing the influence of Mimic-Shift. In order to weaken the effect of averaged gradients from adversaries while strengthening the effect of average gradient from benign users, the proposed method utilizes projection to judge whether a client is a benign user or an adversary client in the setting where exists majority adversaries. The effectiveness of Mimic-shift and Mimic-shift-Par is shown and compared with other kinds of attacks. The performance of the proposed method against attacks is demonstrated with the report of accuracies in several classic benchmarks of Federated learning (FEMNIST, CelebA, and Shakespeare), compared with some baselines designed for defending against standard attacks.

However, there are several obvious weakness: 1) The motivation for the proposed second-stage reweighting is not very clear. It could be better improved by adding a more intuitive example or illustration that the reweighting mechanism is the target choice for defending the mimic-shift attack. 2) The technical novelty of the current presentation is limited, as the reweighting mechanism or filtering is commonly used to defend against poisoning attacks. It is encouraged to add more discussion about the rationality of adopting reweighting in this problem. 3) Regarding adaptivity, you need to define your defense and allow the adversary to know your defense and still show that the defense works. Meanwhile, the setting of AAA21 paper seems more general as a defense. Namely, they allow a number of parties to submit completely malicious data (or models that are completely malicious). Overall, this paper may not be ready for publication at ICLR. The next version must be a strong paper if authors can take comments into consideration.